# LANGUAGE MODEL FOR LARGE-TEXT TRANSMISSION IN NOISY QUANTUM COMMUNICATIONS

## ABSTRACT

Quantum communication has the potential to revolutionise information processing, providing unparalleled security and increased capacity compared to its classical counterpart by using the principles of quantum mechanics. However, the presence of errors poses a significant challenge to realising these advantages. While strategies like quantum error correction and quantum error mitigation have been developed to address these errors, they often come with substantial overhead, hindering the practical transmission of large texts. Here, we introduce an application of machine learning frameworks for natural language processing to enhence the performance of noisy quantum communications, particularly superdense coding. Using bidirectional encoder representations from transformers (BERT), a model known for its capabilities in natural language processing, we demonstrate that language-model-assisted quantum communication protocols can substantially improve the efficiency of large-scale information transfer. This brings us closer to the practical realisation of a quantum internet.

## 1 INTRODUCTION

In our increasingly interconnected world, effective communication is essential for facilitating the exchange of information, fostering collaboration, and driving problem-solving across various domains. It serves as the foundation of a modern society, enabling the development of technological landscapes. Entering the spotlight, quantum communication Bouwmeester et al. (1997); Gisin & Thew (2007); Cozzolino et al. (2019); Xing et al. (2024) is emerging as a revolutionary field that uses the principles of quantum mechanics to achieve unprecedented levels of security and efficiency, signaling a new era in data transmission. Today, quantum information can already be transmitted from satellites to the ground Ren et al. (2017); Liao et al. (2017); Lu et al. (2022); Li et al. (2024a), and quantum networks are being established within cities (Yin et al., 2012; Liu et al., 2024; Knaut et al., 2024). One notable example of the advantages of quantum communication is superdense coding Bennett et al. (1999); Harrow et al. (2004); Barreiro et al. (2008); Hu & et al. (2018); Li et al. (2024b), which enables a single qubit channel to transmit the equivalent of two classical bits. This is made possible through shared entanglement between the sender and receiver, effectively doubling the amount of information transmitted.

The next step in advancing quantum communication involves utilizing the superdense coding protocol as a subroutine for transmitting text. However, two practical issues need to be addressed. The first, which is relatively straightforward, is transforming the text into a bit string that can be used for superdense coding. This can be easily achieved using standard ASCII code. The second, more complex challenge arises from the influence of noise on quantum communications, which can degrade performance. Current strategies, including quantum error correction and quantum error mitigation, have their pros and cons. For instance, quantum error correction Knill & Laflamme (1997); Aoki et al. (2009); Terhal (2015); Krinner et al. (2022); Sivak et al. (2023) can efficiently handle errors but often requires substantial additional resources that may not be available in practice. On the other hand, quantum error mitigation Endo et al. (2018); Cai et al. (2023) necessitates multiple experimental repetitions to minimize the impact of errors, potentially undermining the advantages offered by superdense coding. Given these challenges, is there an alternative protocol that can effectively mitigate noise in quantum communication without requiring extra quantum resources or multiple communication repetitions?

Here, we address this question by introducing a language-model-assisted quantum protocol which benefits from both superdense coding and natural language processing (NLP). In particular, we incorporate BERT, a pre-trained language model recognized for its ability to understand the context and predict corrections in tasks such as sentiment analysis Hoang et al. (2019); Xu et al. (2019); Sousa et al. (2019); Cao et al. (2021); Prottasha et al. (2022) and spell error correction Zhang et al. (2020); Tan et al. (2020); Nguyen et al. (2020), into the superdense coding framework, resulting in what we term post quantum communicaton BERT (PQC-BERT). PQC-BERT incorporates both a word-level repairing module (WLRM) and a sentence-level repairing module (SLRM), significantly enhancing the fidelity of text transmission in quantum communications. We assess the model's performance by comparing the bit error rate, word error rate, and sentence error rate before and after its implementation through numerical experiments, validating its effectiveness in ensuring reliable text communication. This language-model-assisted quantum communication protocol presents a new method for mitigating noise in quantum communication, potentially accelerating the advancement of hybrid classical-quantum networks.

**Related works** Recent advancements in machine learning techniques have revealed substantial potential for enhancing our understanding and manipulation of complex quantum systems. For example, these methods can optimize quantum circuits, simulate quantum phenomena, and identify new quantum algorithms (Carrasquilla, 2020; Melko et al., 2019). In particular, neural networks are highly effective at approximating quantum states and predicting behavior (Carleo & Troyer, 2017; Carrasquilla & Melko, 2017). Additionally, artificial intelligence is revolutionizing quantum chemistry by predicting molecular properties and designing new materials (Rupp et al., 2012; von Lilienfeld et al., 2020). This synergy not only enhances quantum computations but also tackles challenges that classical computing cannot solve (Biamonte et al., 2017; Dunjko & Briegel, 2018). Reinforcement learning has proven effective in optimizing long-distance quantum communication, improving the performance of communication protocols (Wallnöfer et al., 2020). Moreover, quantum computing offers new insights into machine learning problems (Guo et al., 2024; Yu et al., 2023; Cherrat et al., 2024; Tian et al., 2023; Wang et al., 2023). Beyond these developments, NLP plays a crucial role in information processing. Among the techniques in this field, BERT, a pre-trained model derived from Transformers Vaswani et al. (2017), utilizes self-attention mechanisms to analyze language data (Devlin et al., 2019). It achieves outstanding performance in sentiment analysis Hoang et al. (2019); Bello et al. (2023), spam filtering Cao & Lai (2020); Oswald et al. (2022), named entity recognition Hakala & Pyysalo (2019); Chang et al. (2021), question answering systems, reading comprehension Xu et al. (2019), matching tasks, and information retrieval (Lo & Simard, 2019; Peinelt et al., 2020; Feifei et al., 2020). The success of BERT has spurred further advancements, including models like XLNet Yang et al. (2019), RoBERTa Zhuang et al. (2021), and ALBERT (Lan et al., 2019).

## 2 PRELIMINARIES

This section offers a concise overview of fundamental concepts in quantum information theory, including quantum states, gates, channels, entanglement, and standard noise models. For a more detailed introduction to the topic, we recommend consulting Ref. (Nielsen & Chuang, 2010; Wilde, 2013; Watrous, 2018). Readers who are already well-versed in these concepts may opt to skip this section.

**Quantum state** In quantum information and computing, information is encoded in quantum systems through the preparation of quantum states, with the fundamental unit being the quantum bit, or qubit, which represents a two-dimensional quantum system. In contrast to a classical bit, which can only exist in one of two states, 0 or 1, a qubit can exist in a superposition of states, described by a unit vector in the two-dimensional Hilbert space $\mathbb{C}^2$. The state of a qubit is expressed in Dirac notation as $|\psi\rangle = \alpha_0 |0\rangle + \alpha_1 |1\rangle$, where $|0\rangle = [1, 0]^\mathsf{T}$ and $|1\rangle = [0, 1]^\mathsf{T}$ and $|\alpha_0|^2 + |\alpha_1|^2 = 1$. In this work, the notation $^\mathsf{T}$ indicates the transpose, and $[\alpha, \beta]^\mathsf{T}$ denotes the corresponding column vector. The coefficients $\alpha_0$ and $\alpha_1$ satisfy the normalization condition $|\alpha_0|^2 + |\alpha_1|^2 = 1$, which reflects the principle of superposition. In a system composed of $n$ qubits, the quantum state is represented as a normalized vector within the $n$-fold tensor product Hilbert space, denoted as $(\mathbb{C}^2)^{\otimes n}$. A quantum state in a $d$-dimensional Hilbert space is referred to as a qudit, which can be expressed as $\sum_{i=0}^{d-1} \alpha_i |i\rangle$ (Wang et al., 2020).

**Quantum gate** The fundamental operations for manipulating quantum states are referred to as quantum gates. Due to their reversible nature, quantum gates are represented by unitary matrices. The essential quantum operations for qubits consist of the Pauli gates and the Hadamard gate

$$I = \begin{pmatrix} 1 & 0 \\ 0 & 1 \end{pmatrix}, \quad X = \begin{pmatrix} 0 & 1 \\ 1 & 0 \end{pmatrix}, \quad Y = \begin{pmatrix} 0 & -i \\ i & 0 \end{pmatrix}, \quad Z = \begin{pmatrix} 1 & 0 \\ 0 & -1 \end{pmatrix}, \quad H = \frac{1}{\sqrt{2}} \begin{pmatrix} 1 & 1 \\ 1 & -1 \end{pmatrix}. \tag{1}$$

**Quantum measurement** The standard method for extracting or decoding information from a quantum system involves implementing quantum measurements. A widely used one in quantum machine learning Biamonte et al. (2017) and, more generally, variational quantum computing Cerezo et al. (2021); Bharti et al. (2022); Larocca et al. (2024), is measurement in the computational basis. For a qubit represented as $|\psi\rangle = \alpha_0 |0\rangle + \alpha_1 |1\rangle$, measuring the state yields outcomes $0$ or $1$ with probabilities $|\alpha_0|^2$ and $|\alpha_1|^2$, respectively. Similarly, when measuring a qudit in the computational basis, the state collapses to $|i\rangle$ with probability $|\alpha_i|^2$ for $i \in \{0, \ldots, d-1\}$.

**Quantum Entanglement** A fundamental quantum resource that sets quantum theory apart from classical physics is quantum entanglement Horodecki et al. (2009), which enables technologies such as quantum computing and quantum communication. In many applications, maximally entangled states play an essential role. For qudit systems, these states, commonly referred to as generalized Bell states, are given by

$$|\Phi_{zx}\rangle = (Z(z)X(x) \otimes I) |\Phi_{00}\rangle, \quad \forall z, x \in \{0, \ldots, d-1\}. \tag{2}$$

Here, $|\Phi_{00}\rangle := 1/\sqrt{d} \sum_{i=0}^{d-1} |ii\rangle$, and the Heisenberg-Weyl operators are expressed as Khatri & Wilde (2024)

$$Z(z) = \sum_{i=0}^{d-1} e^{\frac{2\pi i k z}{d}} |i\rangle \langle i|, \quad X(x) = \sum_{i=0}^{d-1} |(i+x) \bmod d\rangle \langle i|. \tag{3}$$

For qubits, the following four standard Bell states are identified.

$$|\Phi_{00}\rangle = \frac{1}{\sqrt{2}}(|00\rangle + |11\rangle), \quad |\Phi_{01}\rangle = \frac{1}{\sqrt{2}}(|10\rangle + |01\rangle),$$
$$|\Phi_{10}\rangle = \frac{1}{\sqrt{2}}(|00\rangle - |11\rangle), \quad |\Phi_{11}\rangle = \frac{1}{\sqrt{2}}(|01\rangle - |10\rangle). \tag{4}$$

These generalized Bell states $\Phi_{zx}$ naturally give rise to projective measurements in the form of $\{|\Phi_{zx}\rangle \langle \Phi_{zx}|\}$, resulting in outcomes $zx$ (Schuck et al., 2006).

**Quantum channel** In a closed quantum system, the evolution of the quantum state is reversible and is described by unitary gates. However, in the broader context of open quantum systems Rivas & Huelga (2012); Pollock et al. (2018); Lidar (2019); Milz & Modi (2021); Xiao et al. (2023); Xing et al. (2023), where the environment is taken into account, the quantum dynamics is characterized by completely positive trace-preserving (CPTP) linear maps known as quantum channels. Mathematically, a quantum channel $\mathcal{E}$ can be decomposed into the following form using Kraus operators $K_i$

$$\mathcal{E}(\cdot) = \sum_i K_i \cdot K_i^\dagger, \tag{5}$$

where the operators satisfy the completeness condition, namely $\sum_i K_i^\dagger K_i = I$, with $\dagger$ denoting the Hermitian adjoint. The Kraus decompositions for various qubit noise models Gottesman (1998); Lidar & Brun (2013); Gottesman (2024) are listed in Table 1.

The bit flip error, as previously discussed, is also a typical error in classical information theory, where bits can be flipped with a certain probability $\lambda$ during transmission. Alongside the qubit case, we will perform numerical experiments for the qudit case with $d = 4$ to examine the bit flip error in the following sections. Accordingly, we also present the Kraus operators of bit flip channel specific to the 4-dimensional case Khatri & Wilde (2024) in Table 1.

Table 1: **Noise Models.** The first four rows list commonly used noise models for qubits, while the last row represents the bit-flip error for qudits with $d = 4$. The parameter $\lambda$ denotes the noise strength, representing the magnitude of noise affecting the communication process.

| Noise Models | Kraus Operators |
|:---:|:---:|
| Bit Flip Channel | $K_0 = \sqrt{1-\lambda}I, \quad K_1 = \sqrt{\lambda}X$ |
| Phase Flip Channel | $K_0 = \sqrt{1-\lambda}I, \quad K_1 = \sqrt{\lambda}Z$ |
| Depolarizing Channel | $K_0 = \sqrt{1-3\lambda/4}I, \quad K_1 = \sqrt{\lambda/4}X,$ $K_2 = \sqrt{\lambda/4}Y, \quad K_3 = \sqrt{\lambda/4}Z$ |
| Amplitude Damping Channel | $K_0 = \begin{pmatrix} 1 & 0 \\ 0 & \sqrt{1-\lambda} \end{pmatrix}, K_1 = \begin{pmatrix} 0 & \sqrt{\lambda} \\ 0 & 0 \end{pmatrix}$ |
| Qudit Bit Flip Channel($d = 4$) | $K_0 = \sqrt{1-\lambda}I, \quad K_1 = \sqrt{\lambda/3}X(1),$ $K_2 = \sqrt{\lambda/3}X(2), \quad K_3 = \sqrt{\lambda/3}X(3).$ |

## 3 METHODS

In this section, we integrate superdense coding with machine-learning-based NLP to explore the language-model-assisted quantum communications. This modifies the traditional process of transmitting information – encoding $\rightarrow$ noise $\rightarrow$ decoding – into a more robust pipeline: pre-encoding $\rightarrow$ encoding $\rightarrow$ noise $\rightarrow$ decoding $\rightarrow$ post-decoding, as illustrated in Figure. 1(a). Specifically, superdense coding offers an efficient way for transmitting classical information using quantum communication, with the potential to double the transmission capacity compared to classical approaches. However, superdense coding is typically limited to bit string inputs. To enable large-scale text transmission, the text is first converted into 8-bit ASCII code in the pre-encoding process, making it compatible with superdense coding. The encoded bit strings are then transmitted, though noise during the process can degrade performance. After decoding, machine learning-based NLP techniques are employed in a post-processing phase, helping to recover and refine the transmitted information. This approach mitigates the impact of noise and enhances the overall reliability and efficiency of quantum communications.

### 3.1 QUANTUM SUPERDENSE CODING

Quantum superdense coding is a protocol that enables the transmission of classical bits using fewer qubits, provided that the sender and receiver share a pre-established entangled resource. In the qubit case, if the parties pre-share a maximally entangled state $[qq]$, and the sender has access to a noiseless qubit channel $[q \rightarrow q]$ to transmit a qubit, it is possible to communicate 2 classical bits of information. In other words, this simulates the process of $2[c \rightarrow c]$.

$$[qq] + [q \rightarrow q] \geqslant 2[c \rightarrow c]. \tag{6}$$

The formal procedure is outlined as follows:

1. **Entanglement Distribution.** A third party prepared the maximally entangled state $|\Phi_{00}\rangle$ and distributed it to the sender, Alice, and the receiver, Bob.

2. **Encoding.** Alice encodes two bits of classical information $zx \in \{0,1\}^2$ into one of the Bell states $|\Phi_{zx}\rangle$ by applying Pauli operations $Z(z)X(x)$ to her qubit.

3. **Noisy Communication.** Alice then sends her qubit to Bob through the noisy quantum channel $\mathcal{E}$.

4. **Decoding.** Upon receiving the qubit from Alice, Bob performs a Bell measurement. Based on the measurement outcome, Bob decodes the classical information $zx$ that Alice encoded.

A schematic representation of the superdense coding protocol is provided in Figure. 1(b). By substituting the Pauli gates with Heisenberg-Weyl operators, we can readily extend the protocol from the qubit case to the general qudit case.

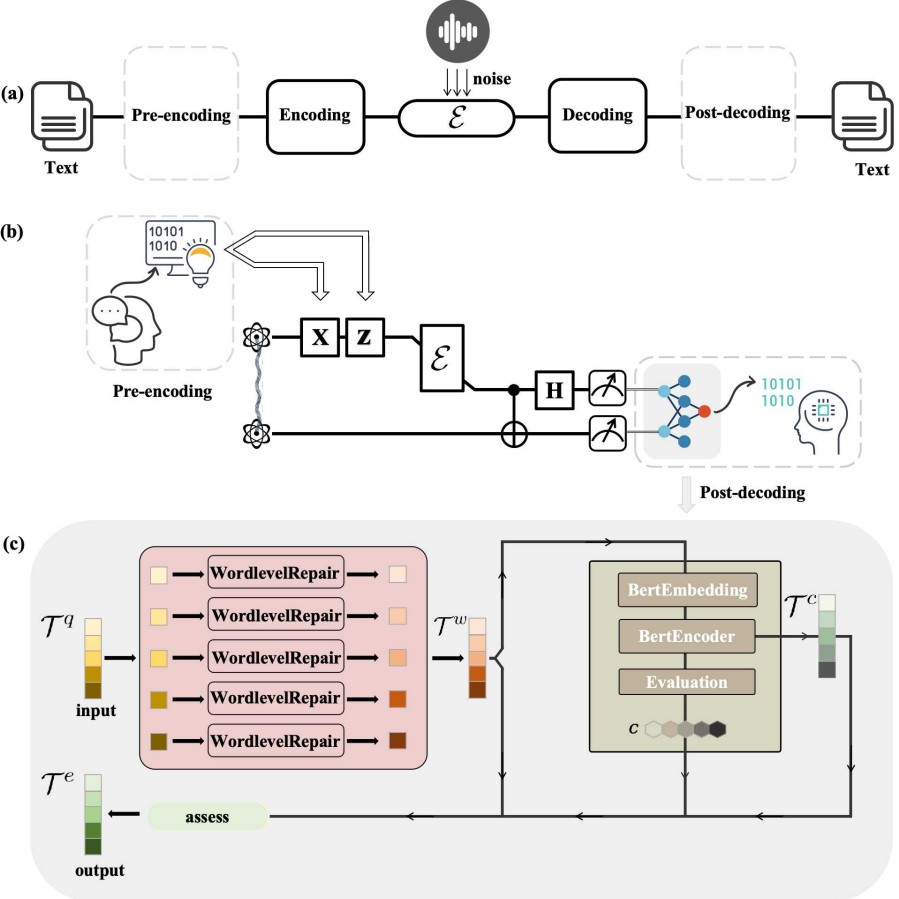

Figure 1: **Language-Model-Assisted Quantum Communication**. Figure 1(a) shows the outline of our communication protocol designed for large-text transmission tasks. Figure 1(b) explicitly illustrates the standard superdense coding. In addition to superdense coding, we encapsulate it with a pre-encoding process and a post-decoding process based on a language model. Figure 1(c) further details the structure of our post-decoding component, including the word-level repairing module (WLRM) and the sentence-level repairing module (SLRM).

## 3.2 PRE-ENCODING PROCESSING

To transmit text using a subroutine that incorporates superdense coding, we first convert the characters into a bit string using the standard 8-bit ASCII code. We refer to this process as "ASCII encoding" and denote it by $\mathcal{A}$. This encoded bit string is then processed through the superdense coding protocol. We denote the original content that sender Alice wishes to transmit to receiver Bob as $\mathcal{T} = [w_1, \cdots, w_n]$, where $w_i$ ($i \in [n] := \{1, \ldots, n\}$) represents a word within the text. After transmitting the bit string through the superdense coding protocol, where a noisy channel $\mathcal{E}$ has occurred, we obtain an output bit string. To recover the transmitted message, we apply the inverse mapping $\mathcal{A}^{-1}$ to convert the output bit string back into the message $\mathcal{T}^q = [w_1^q, \cdots, w_n^q]$. Similar to the original content $\mathcal{T}$, each $w_i^q$ signifies a word, with $n$ representing the total number of words. Additionally, each word $w_i^q$ can be expressed as $[l_{i1}^q, l_{i2}^q, \cdots, l_{im}^q]$, consisting of $m$ letters.

### 3.3 Post-decoding processing

Our newly introduced post-decoding process comprises two main components: word-level repairing module (WLRM) and sentence-level repairing module (SLRM). The first component utilizes a predefined dictionary, denoted as Dict, to rectify misspelled words at the word level. The second component employs a language model to tackle more intricate semantic errors at the sentence level.

#### 3.3.1 Word-level repairing module (WLRM)

We begin by introducing our word-level repairing module (WLRM). For a given word $w_i^q$, derived from superdense coding and subsequently mapped through the inverse mapping $\mathcal{A}^{-1}$ of the "ASCII encoding", we assess its validity against the shared dictionary Dict. If $w_i^q$ is found in Dict, we redefine $w_i^q$ as $w_i^w$ and proceed to the next word $w_{i+1}^q$. Otherwise, the word is identified as misspelled, prompting us to select candidate corrections $w_i^w$ from Dict based on the following measure

$$\Delta(w, w_i^q) := \sum_{j=1}^{m} \text{Hamming}\left(\mathcal{A}(l_j), \mathcal{A}(l_{ij}^q)\right), \tag{7}$$

where $\text{Hamming}(\cdot, \cdot)$ denotes the Hamming distance, and $l_j$ represents the $j$-th letter of the word $w$. Based on this distance, our objective is to find a word in Dict that minimizes the aforementioned distance, which we denote as

$$w_i^w := \underset{w \in \text{Dict}}{\arg\min} \Delta(w, w_i^q). \tag{8}$$

By repeatedly applying this process, we obtain a word-level repaired text $\mathcal{T}^w = [w_1^w, \cdots, w_n^w]$. This text $\mathcal{T}^w$ is subsequently fed into the next sentence-level repairing module (SLRM). It is worth noting that using spell-corrected text as tokenized input can enhance the correction accuracy of the BERT-based module.

#### 3.3.2 Sentence-level repairing module (SLRM)

Our sentence-level repairing module (SLRM) is constructed using BERT and comprises two interconnected networks: the correction network and the evaluation network. The correction network utilizes BERT Devlin et al. (2019) to analyze the linguistic context effectively. However, we have observed that BERT tends to recommend changes even when the original words are correct. To counter this behavior, we incorporate an evaluation network that assesses the proposed edits from the correction network, making decisions about whether to accept or reject them.

#### 3.3.3 Correction network

Our correction network is a sequential multi-class labeling model based on BERT Devlin et al. (2019), designed to take corrected text $\mathcal{T}^w$ as input and generate contextually accurate sentences $\mathcal{T}^c$. The model consists of 12 stacked identical blocks, each comprising a multi-head self-attention layer followed by a feedforward network. Every word $w_i^w$ in the input $\mathcal{T}^w$ is transformed into an embedding vector, combined with positional embeddings to encode word order, and marked with special tokens [CLS] at the start and [SEP] at the end. The multi-head self-attention mechanism, described in Vaswani et al. (2017), allows the model to capture global dependencies between words in the sequence, irrespective of their distance. The attention function is computed simultaneously on the combined sets of queries $Q$, keys $K$, and values $V$ as follows

$$\text{Attention}(Q, K, V) := \text{softmax}\left(\frac{QK^T}{\sqrt{d_k}}\right)V \tag{9}$$

where $d_k$ is the dimension of the $Q$. Each attention head, i.e., $\text{head}_h$, learns distinct aspects of these relationships, expressed as

$$\text{MultiHead}(Q, K, V) := \text{Concatenation}(\text{head}_1, \text{head}_2, \ldots, \text{head}_h)W^O \tag{10}$$

where

$$\text{head}_i := \text{Attention}(QW_i^Q, KW_i^K, VW_i^V) \tag{11}$$

and $W^O, W_i^Q, W_i^Q, W_i^Q$ are the weight matrixs parameters obtained by learning.

Following the self-attention sublayer, a position-wise feed-forward network (FFN) is applied independently, whose input is the output $x$ from the previous sublayer. The process of current layer can be formulated as

$$\text{FFN}(x) := \max(0, xW_1 + b_1)W_2 + b_2 \tag{12}$$

where $W_1, W_2$ stand for the learnable weight matrices, and $b_1, b_2$ are the biases. Outputs from each sublayer are followed by residual connections and layer normalization, which respectively mitigate the vanishing gradient problem and stabilize training in deep networks.

Let the output of BERT be denoted as $P^c = [p_1^c, p_2^c, \cdots, p_n^c]$, where $p_i^c$ is a probability vector. Each element in $p_i^c$ corresponds to the probability of a specific dictionary word appearing in the $i$-th position of the text. For example, if the word 'quantum' has the highest probability among all elements in $p_1^c$, the first word of the text will be 'quantum.' To construct the full text, we select the word with the highest probability for each position, producing the text $\mathcal{T}^c = [w_1^c, \cdots, w_n^c]$.

### 3.3.4 EVALUATION NETWORK

Our evaluation network will take $\mathcal{T}^c$ as input and produce a vector of confidence scores $c = [c_1, c_2, \ldots, c_n]$. If the score for the $i$-th position, namely $c_i$, is close to 0, we conclude that the output from BERT is over-corrected; thus, we should select $w_i^w$ from the word-level repairing module (WLRM) as the $i$-th word in the text. Conversely, if the score is close to 1, we trust the correction provided by BERT, indicating that the sentence-level repairing module (SLRM) is necessary, and we take the $i$-th word from the text as $w_i^c$. We denote the resulting text from this process as $\mathcal{T}^e$.

### 3.4 LEARNING

The learning phase is focused on the SLRM component. The training dataset consists of input-output pairs $(\mathcal{T}^w, \mathcal{T})$, where $\mathcal{T}^w$ denotes the texts corrected by the WLRM, and $\mathcal{T}$ represents the original texts that the sender, Alice, intends to transmit to the receiver, Bob. The learning objective is composed of two key components: correction and evaluation. To address these, we formulate two separate loss functions – one for the correction network and another for the evaluation network. The loss function $\mathcal{L}_c$ for the correction network is defined as

$$\mathcal{L}_c := -\sum_{i=1}^{n} \log P(w_i^c | \mathcal{T}^w), \tag{13}$$

where $P(w_i^c | \mathcal{T}^w)$ denotes the probability of selecting the word $w_i^c$ given the input $\mathcal{T}^w$. The loss function $\mathcal{L}_e$ for the evaluation network is based on Focal Loss (Ross & Dollár, 2017). Given that incorrect words in $\mathcal{T}^q$ represent a relatively small portion of our dataset, we employ a modified Focal Loss to address the challenge of class imbalance. Using the word-level repaired text $\mathcal{T}^w$, we create a boolean flag array $q = [q_1, q_2, \ldots, q_n]$ to indicate whether $w_i^w = w_i$, i.e., $q_i := \mathbf{1}\{w_i^w = w_i\}$. Meanwhile, the modified focal loss function is defined as

$$\mathcal{L}_e := -\sum_{i=1}^{n} \alpha(1 - f_i)^\gamma \log(f_i + \epsilon), \tag{14}$$

where $f_i = c_i$ when $q_i = 1$ (indicating that the word should be replaced), and $f_i = 1 - c_i$ otherwise (indicating that the word should not be replaced). The parameters $\alpha$, $\gamma$, and $\epsilon$ are employed to fine-tune the Focal Loss function, thereby enhancing its effectiveness in handling the imbalanced dataset.

Ultimately, we take the linear combination of Eq. 13 and Eq. 14 as the final loss function for training

$$\mathcal{L} := \theta \cdot \mathcal{L}_c + (1 - \theta) \cdot \mathcal{L}_e, \quad \theta \in [0, 1]. \tag{15}$$

The remaining hyperparameters and learning rates were established based on prior work (Quijano et al., 2021). We selected the Adam optimizer and implemented slight adjustments to enhance its performance.

## 4 NUMERICAL EXPERIMENTS

In this section, we present numerical experiments to demonstrate the advantages of our language-model-assisted communication protocols, emphasizing the role of machine learning, particularly NLP, in enhancing communication. We compare the performance of bit and qubit communication in the presence of bit-flip errors, finding that quantum communication consistently surpasses classical methods in terms of sentence accuracy, recall, precision, and F1 score. Furthermore, our evaluation of the language-model-assisted communication protocols shows that the PQC-BERT module effectively mitigates noise, significantly reducing the bit error rate, word error rate, and sentence error rate. These results highlight the superior performance of language-model-assisted quantum communication.

### 4.1 DATASETS

Our numerical experiments are based on modified versions of two established datasets: Flickr-8k Text Captioning Dataset (Flickr-8k) Hodosh et al. (2013) and Corpus of Linguistic Acceptability (CoLA) (Warstadt et al., 2019). The Flickr-8k dataset includes $8,000$ images, each paired with $5$ distinct captions that effectively describe the key entities and events within the images. In contrast, CoLA serves as a benchmark for single-sentence classification, containing sentences sourced from 23 linguistic publications, expertly annotated for acceptability by their original authors. To assess the performance of our models on text transmission tasks, we manually extracted $12,459$ corrected English sentences from Flickr-8k and $9,078$ from CoLA, resulting in two new datasets: Mini-Flickr and Mini-CoLA.

### 4.2 SETTING

Given a noisy channel $\mathcal{E}$ and ideal text $\mathcal{T}$. we first simulate the effects of $\mathcal{E}$ and then apply WLRM to generate the modified text $\mathcal{T}^w$ through word-level correction. During training, we use $\mathcal{T}^w$ as the input and $\mathcal{T}$ as the output to train our neural network. For testing, we evaluate the similarity between the original ideal text $\mathcal{T}$ (before it passes through the noisy channel $\mathcal{E}$) and the text produced after the post-decoding process. This evaluation encompasses the entire language-model-assisted communication protocol. To validate our approach, we conduct independent and replicated experiments to demonstrate the average performance of our method.

### 4.3 RESULTS

We assess the performance of our models using standard metrics to evaluate language models: **Accuracy**, **Recall**, **Precision**, and **F1-score**. As shown in Table 2, we demonstrate the performance of PQC-BERT under a bit-flip noise parameter of $0.01$ in both quantum and classical channels. The results reveal that PQC-BERT performs effectively, demonstrating its robustness in handling noise.

Table 2: **Performance Analysis of PQC-BERT in Classical and Quantum Communications.** PQC-BERT was assessed on classical and quantum channels (qubit) with a fixed noise parameter of 0.01 for bit flip error. The results indicate that our SLRM module performed well in the text repair task for both cases.

| Dataset | Channel | Accuracy | Precision | Recall | F1-Score |
|---|---|---|---|---|---|
| Mini-Flickr | Classical | 0.6336 | 0.6583 | 0.6167 | 0.7045 |
| | Quantum | **0.7642** | **0.7971** | **0.7491** | **0.7725** |
| Mini-CoLA | Classical | 0.5104 | 0.7108 | 0.5903 | 0.5172 |
| | Quantum | **0.7401** | **0.7971** | **0.6611** | **0.6765** |

To further demonstrate the advantages of the entire language-model-assisted communication protocol, we evaluate its performance across various noise parameters and high-dimensional qudit channels with $d = 4$ in Table 3. Our analysis focuses on three key metrics:

1. **Bit Error Rate.** This metric quantifies the ratio of erroneous bits to the total number of bits transmitted. It is affected by the channel's noise parameters, providing valuable insights into the system's sensitivity to noise, its interference suppression capabilities, and its overall communication capacity – making it a crucial metric in information theory.

2. **Word Error Rate.** This metric assesses the proportion of errors within transmitted data blocks, defined as the number of erroneous words divided by the total number of transmitted words. It effectively evaluates the quality of transmission at the word level.

3. **Sentence Error Rate.** This metric measures the performance of our text communication protocol by calculating the ratio of erroneous sentences to the total number of sentences transmitted. It provides valuable insight into the protocol's accuracy at the sentence level.

Table 3: **Performance Analysis of Language-Model-Assisted Communication Protocols Under Varying Noise Conditions.** We assess the quality of text transmission across classical bit communication, quantum qubit communication, and quantum qudit communication with $d = 4$ under various noise parameters of the bit-flip channel and using different datasets. For each metric, we present two columns: one showing the error rate before the application of the PQC-BERT module and the other reflecting the error rate after its application, with the latter presented in bold font. This bold formatting highlights the effectiveness of our protocols.

(a) Language-model-assisted classical protocol

| Dataset | Noise.Para | Bit Error Rate | | Word Error Rate | | Sentence Error Rate | |
|---|---|---|---|---|---|---|---|
| Mini-Flickr | 0.01 | 0.0101 | **0.0086** | 0.2426 | **0.1979** | 0.9316 | **0.3664** |
| | 0.005 | 0.0050 | **0.0040** | 0.1271 | **0.0269** | 0.7643 | **0.2239** |
| | 0.001 | 0.0012 | **0.0010** | 0.0287 | **0.0239** | 0.2718 | **0.1361** |
| Mini-CoLA | 0.01 | 0.0102 | **0.0100** | 0.2911 | **0.2146** | 0.9071 | **0.4896** |
| | 0.005 | 0.0050 | **0.0045** | 0.1576 | **0.0364** | 0.7173 | **0.2676** |
| | 0.001 | 0.0010 | **0.0009** | 0.0376 | **0.0324** | 0.2486 | **0.1963** |

(b) Language-model-assisted quantum protocol (qubit)

| Dataset | Noise.Para | Bit Error Rate | | Word Error Rate | | Sentence Error Rate | |
|---|---|---|---|---|---|---|---|
| Mini-Flickr | 0.01 | 0.0049 | **0.0018** | 0.1301 | **0.0325** | 0.7685 | **0.2358** |
| | 0.005 | 0.0024 | **0.0016** | 0.0706 | **0.0091** | 0.5376 | **0.1054** |
| | 0.001 | 0.0004 | **0.0003** | 0.0141 | **0.0054** | 0.1755 | **0.0542** |
| Mini-CoLA | 0.01 | 0.0062 | **0.0211** | 0.1646 | **0.0577** | 0.7306 | **0.2599** |
| | 0.005 | 0.0031 | **0.0019** | 0.0859 | **0.0431** | 0.4858 | **0.1954** |
| | 0.001 | 0.0005 | **0.0004** | 0.0205 | **0.0184** | 0.1348 | **0.0665** |

(c) Language-model-assisted quantum protocol (qudit)

| Dataset | Noise.Para | Bit Error Rate | | Word Error Rate | | Sentence Error Rate | |
|---|---|---|---|---|---|---|---|
| Mini-Flickr | 0.01 | 0.0034 | **0.0016** | 0.0686 | **0.0036** | 0.5313 | **0.1349** |
| | 0.005 | 0.0017 | **0.0013** | 0.0349 | **0.0027** | 0.3027 | **0.0914** |
| | 0.001 | 0.0003 | **0.0001** | 0.0081 | **0.0023** | 0.0918 | **0.0305** |
| Mini-CoLA | 0.01 | 0.0037 | **0.0023** | 0.0945 | **0.0605** | 0.4877 | **0.1955** |
| | 0.005 | 0.0016 | **0.0013** | 0.0436 | **0.0358** | 0.2752 | **0.1049** |
| | 0.001 | 0.0003 | **0.0002** | 0.0114 | **0.0109** | 0.0968 | **0.0565** |

With a fixed noise parameter, quantum communication protocols based on either qubits or qudits ($d = 4$) outperform their classical counterparts. This advantage becomes increasingly pronounced as the dimensionality of the qudit increases. For instance, when the noise parameter is set at $0.001$,

language-model-assisted classical communication achieves a sentence error rate of $13.61\%$. In contrast, the sentence error rate for language-model-assisted quantum communication using qubits is reduced to $5.42\%$. Furthermore, when qubits are replaced by qudits ($d = 4$), the performance improves further, yielding a sentence error rate of only $3.05\%$. These findings demonstrate that quantum dense coding not only offers high capacity but also exhibits greater robustness against noise.

Our numerical experiments confirm the necessity of employing machine learning techniques, such as language models, in quantum communication. For instance, when using a bit flip channel with $\lambda = 0.01$ to transmit qubit states, we observe a significant sentence error rate – $76.85\%$ for Mini-Flickr and $73.06\%$ for Mini-CoLA – despite the low noise strength. These high error rates render the sentences largely unreadable. However, with our language model-assisted quantum communication protocol, specifically after applying the PQC-BERT module, we successfully reduce the sentence error rates to $23.58\%$ for Mini-Flickr and $25.99\%$ for Mini-CoLA. This improvement allows the receiver, Bob, to start reading the text sent to him. In addition to the bit flip error analyzed here, we have also conducted experiments with other quantum noise models. Due to the absence of classical counterparts for these models, we are unable to make comparisons between language model-assisted classical and quantum communications, and thus, we do not present those data here.

## 5 CONCLUSION

Quantum communications offer significant advantages over classical protocols, particularly in capacity and privacy. However, these benefits are often undermined by noise, which is nearly unavoidable in practical applications. In this work, we leverage the power of language models in machine learning to bolster the performance of a key quantum communication protocol – quantum superdense coding – in the presence of noise. We present PQC-BERT, a model that extends conventional superdense coding to enable the transmission of text while effectively mitigating errors that arise during quantum communication. Distinct from conventional quantum error correction techniques, our method protects information without the need for additional systems. Furthermore, unlike typical quantum error mitigation strategies, PQC-BERT operates without requiring extra samples to address noise effects. This makes our approach both resource-efficient and easy to implement, highlighting the remarkable potential of classical machine learning techniques in advancing quantum communications.

The fusion of classical machine learning with quantum superdense coding also opens up new avenues for exploration and raises intriguing questions for future research: (1) Superdense coding is typically used to transmit classical bit strings through quantum communication, demonstrating its advantage in conveying more information per channel use. A key challenge now lies in extending the protocol to support more complex data types, such as images and audio. Given the distinct structure and features of such data, more advanced pre-encoding schemes – beyond the "ASCII encoding" – may be needed to fully optimize the transmission process. Developing these schemes could unlock even broader applications of quantum communication protocols in the future. (2) While our focus has been on language models and superdense coding, quantum communication encompasses a wider array of tasks. As we move towards building a quantum internet, there is exciting potential to integrate artificial intelligence with other key protocols, such as quantum teleportation and quantum repeaters, to further push the boundaries of this field. (3) In NLP domain, various models have emerged following the BERT architecture. Will these subsequent models outperform BERT in the post-decoding phase of superdense coding? Could different language models reveal distinct advantages for different noise models? Our numerical experiments are based on processed datasets, Mini-Flickr and Mini-CoLA, masked with simulated quantum noise. It will be beneficial to further evaluate the performance of language-model-assisted quantum communications in real physical systems, such as photonics. However, exploring this aspect falls beyond the scope of the current work and will be addressed in future research.

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
