# OpenReview forum: "Language Model for Large-Text Transmission in Noisy Quantum Communications"
_ICLR.cc/2025/Conference — ICLR 2025 Conference Withdrawn Submission_

### Official Review · Reviewer_DqsF · 2024-10-27

**Soundness:** 2
**Presentation:** 1
**Contribution:** 1
**Rating:** 1
**Confidence:** 3

**Summary:**

The authors proposed using language models in quantum noisy communications. Two levels of corrections were introduced, one on the world level and one on the sentence level. The authors simulated and showed that it is beneficial to use language models for correction.

**Strengths:**

Using machine learning techniques in quantum communication may deserve some research attention.

**Weaknesses:**

Overall I feel the quantum noisy communication is nothing special in this setting, and the method taken is almost naive in some sense. There have been a lot of work using transformers and/or deep neural networks in communication systems in more sophisticated manners, and the authors essentially used a naive communication strategy without data compression or error-correcting codes, and directly applied neural networks at the decoder to resolve the errors. There are several key issues:

1. Reasonable baselines. As mentioned by the authors, there have been many works using quantum error control codes. Why are they not compared?
2. Lack of data compression. It is well known that data compression can reduce the number of bits to represent the text. Why it is not adopted here?

I found the authors spent a large amount of space discussing quantum information and transformer structures, however, almost none is really needed in the development of the corresponding design and evaluation. We can easily replace the quantum channel with a classical channel, and replace the transformers with LSTMs, and it would make little difference in the design.

**Questions:**

1. Why are methods using quantum error control codes not considered and compared?
2. Why is data compression not compared?
3. What aspects make the design special for quantum communication?

---

### Official Review · Reviewer_D6S1 · 2024-10-30

**Soundness:** 2
**Presentation:** 2
**Contribution:** 2
**Rating:** 3
**Confidence:** 4

**Summary:**

The paper proposes a language-model-assisted quantum communication protocol, PQC-BERT, to enhance the performance of quantum superdense coding for transmitting text in the presence of noise. The protocol incorporates a pre-encoding process and a post-decoding process based on the BERT language model, which includes word-level and sentence-level repairing modules. Numerical experiments on modified Flickr-8k and CoLA datasets demonstrate the effectiveness under varying noise conditions.

**Strengths:**

The authors propose an interesting approach that integrates classical machine learning techniques, specifically the BERT language model, with quantum superdense coding. The paper is well presented.

**Weaknesses:**

- The paper does not provide a clear and compelling justification for the necessity of quantum communication in the context of the proposed language-model-assisted protocol.
- The paper only considers single-system quantum noise, specifically the bit-flip channel. However, in practical quantum systems, there are other types of errors, such as crosstalk errors and measurement errors, which are not addressed in the current work.
- The paper does not provide sufficient details on how classical text data is efficiently encoded into quantum states for quantum transmission. A more comprehensive discussion on the encoding and readout of classical data, as well as the resources needed, in the quantum setting is necessary to assess the practicality of the proposed approach.
- The numerical experiments are limited to specific datasets.
- While the numerical experiments claim the effectiveness of PQC-BERT in reducing error rates compared to classical communication, the paper lacks a rigorous theoretical analysis of the claimed advantage.
- The significance and soundness of the contributions are overall not sufficient for ICLR.

**Questions:**

Please refer to the Weaknesses.

---

### Official Review · Reviewer_4Lav · 2024-11-03

**Soundness:** 2
**Presentation:** 3
**Contribution:** 2
**Rating:** 3
**Confidence:** 4

**Summary:**

The paper proposes to combine dense quantum coding (a protocol for communicating classical information via a quantum channel) with classical methods for correcting errors in texts to obtain quantum communication of texts in natural language with a better accuracy.

The paper proposes a machine learning architecture for correcting errors in a text communicated via dense coding and compares the performance between:
- classical noisy communication + machine learning postprocessing;
- quantum dense coding over a noisy channel + machine learning postprocessing.
It finds better performance in the second case.

**Strengths:**

Original combination of quantum communication and classical text processing.

**Weaknesses:**

My main issue with this paper is that quantum dense coding does not provide actual savings in communication. To communicate two classical bits via one qubit, one first has to establish an entangled pair between the two parties. Establishing this pair requires communicating a qubit. Thus, the total communication is actually two qubits: one to establish the entangled pair and one to communicate the two classical bits.

Quantum dense coding is interesting from a conceptual perspective, as it replaces two bits of data-dependent computation with two qubits, one of which is data-independent (the one that is used to establish the entangled) and only one is data-independent. However, it does not give an advantage in the overall communication and, therefore, is very unlikely to be used for communicating text or images.

A second issue is the error model which assumes that the initial entangled state is noiseless but the input-dependent qubit of the communication is noisy. As both the entangled state creation and the later qubit communication happen via a quantum communication channel, it would be logical to assume either that both steps are error-corrected or both steps are noisy. Assuming that one step is noiseless in the quantum case can make the comparison between quantum and classical protocols unfair. (On a high level, this corresponds to decreasing the effective error rate by a half.)

A simple alternative would be to subject both qubits of the entangled pair to the same error channel \Epsilon.

**Questions:**

None.

---

### Note · Authors · 2024-11-18

I have read and agree with the venue's withdrawal policy on behalf of myself and my co-authors.